# RNA-binding protein Mub1 and the nuclear RNA exosome act to fine-tune environmental stress response

Adrien Birot[1], Krzysztof Kus[1], Emily Priest[1], Ahmad Al Alwash[1], Alfredo Castello[1,2], Shabaz Mohammed[1], Lidia Vasiljeva[1], Cornelia Kilchert[3]

The nuclear RNA exosome plays a key role in controlling the levels of multiple protein-coding and non-coding RNAs. Recruitment of the exosome to specific RNA substrates is mediated by RNA-binding co-factors. The transient interaction between co-factors and the exosome as well as the rapid decay of RNA substrates make identification of exosome co-factors challenging. Here, we use comparative poly(A)+ RNA interactome capture in fission yeast expressing three different mutants of the exosome to identify proteins that interact with poly(A)+ RNA in an exosome-dependent manner. Our analyses identify multiple RNA-binding proteins whose association with RNA is altered in exosome mutants, including the zinc-finger protein Mub1. Mub1 is required to maintain the levels of a subset of exosome RNA substrates including mRNAs encoding for stress-responsive proteins. Removal of the zinc-finger domain leads to loss of RNA suppression under non-stressed conditions, altered expression of heat shock genes in response to stress, and reduced growth at elevated temperature. These findings highlight the importance of exosome-dependent mRNA degradation in buffering gene expression networks to mediate cellular adaptation to stress.

## Introduction

Regulation of RNA maturation and degradation is crucial to accurate gene expression (Kilchert et al, 2016). The nucleolytic RNA exosome complex is involved in the biogenesis of multiple types of transcripts produced by RNA polymerases I, II, and III (Pol I, II, and III) (Mitchell et al, 1997; Allmang et al, 1999; Isken & Maquat, 2007; Kiss & Andrulis, 2010; Gudipati et al, 2012; Schneider et al, 2012a; Chlebowski et al, 2013; Szczepińska et al, 2015). The nuclear RNA exosome functions in RNA processing (3′ end trimming) of stable non-coding (nc) RNA species such as small nuclear and nucleolar RNAs (snRNAs and snoRNAs), ribosomal RNAs (rRNAs), and telomerase RNA, as well as in quality control, where it degrades incorrectly processed

mRNAs and ncRNAs (Mitchell et al, 1997; Allmang et al, 1999; Isken & Maquat, 2007; Kiss & Andrulis, 2010; Gudipati et al, 2012; Schneider et al, 2012a; Chlebowski et al, 2013; Coy et al, 2013; Szczepińska et al, 2015; Tseng et al, 2018). The exosome also degrades short-lived nc transcripts produced by Pol II such as promoter upstream transcripts (PROMPTs), enhancer RNAs, and products of wide-spread premature transcription termination in higher eukaryotes, and cryptic unstable transcripts (CUTs) in yeast (Wyers et al, 2005; Preker et al, 2008; Vasiljeva et al, 2008a; Neil et al, 2009; Zhou et al, 2015; Tatomer et al, 2019; Liu et al, 2020). Recent studies have also demonstrated that the exosome not only removes unprocessed mRNAs but also promotes proper mRNA processing because exosome mutants show splicing and mRNA 3′ end–processing defects (Nag & Steitz, 2012; Castelnuovo et al, 2014; Lemay et al, 2014; Shah et al, 2014; Kilchert et al, 2015). Finally, the exosome regulates the levels of specific mRNAs in response to environmental changes and is an important player in executing specific gene expression programmes during development (Harigaya et al, 2006; Chen et al, 2011; Gudipati et al, 2012; Boczonadi et al, 2014; Kilchert et al, 2015; Yang et al, 2020). In various human cell culture models, the activity of the exosome complex was shown to prevent cellular differentiation and maintain cells in an un-differentiated state by suppressing the expression of developmental regulators (McIver et al, 2014, 2016; Lloret-Llinares et al, 2018; Belair et al, 2019). Mutations in the nuclear exosome lead to severe neurodegenerative diseases in humans, such as spinal muscular atrophy and pontocerebellar hypoplasia (Wan et al, 2012; Boczonadi et al, 2014; Yang et al, 2020).

The nuclear RNA exosome is a 3′–5′ exonuclease complex that consists of a nine-protein core (EXO-9) and two catalytic subunits, Rrp6 (EXOSC10) and Dis3/Rrp44 (hDIS3). EXO-9 forms a double-layered barrel-like structure that comprises six ribonuclease (RNase) PH-like proteins (Rrp41, Rrp42, Rrp43, Rrp45, Rrp46, and Mtr3) and three S1/K homology (KH) "cap" proteins (Rrp4, Rrp40, and Csl4) (Makino et al, 2013). The two catalytic subunits occupy opposite ends of EXO-9 to constitute EXO-11 (Tomecki et al, 2010; Januszyk & Lima, 2014). Rrp6 is located at the top of the S1/KH cap ring where RNA substrates enter the channel formed by the exo-some core, whereas Dis3 is found at the bottom of EXO-9 near the channel exit. Both Rrp6 and Dis3 are 3′–5′ exonucleases, but the

[1]Department of Biochemistry, University of Oxford, Oxford, UK  [2]MRC-University of Glasgow Centre for Virus Research, Glasgow, UK  [3]Institute of Biochemistry, Justus-Liebig University Giessen, Giessen, Germany

Correspondence: cornelia.kilchert@chemie.bio.uni-giessen.de; lidia.vasilieva@bioch.ox.ac.uk

latter also has endonucleolytic activity (Chlebowski et al, 2013). In yeast, Rrp6 is restricted to the nucleus, whereas Dis3 is found in both the nucleus and the cytoplasm (Shiomi et al, 1998; Tomecki & Dziembowski, 2010).

The conserved Ski2-like helicase Mtr4 is essential for RNA degradation by the nuclear exosome (Houseley & Tollervey, 2009). In the fission yeast, *Schizosaccharomyces pombe* (*S. pombe*), Mtr4 shares its function with the highly homologous Mtr4-like helicase Mtl1 (Lee et al, 2013). Mtr4/Mtl1 interact with the exosome cap structure and are thought to play a central role in exosome recruitment to substrate RNAs and the facilitation of RNA degradation through substrate unwinding (Kilchert et al, 2016; Weick et al, 2018; Lingaraju et al, 2019; Weick & Lima, 2020). In addition to the exosome core, Mtr4/Mtl1 co-purify with RBPs involved in substrate recognition. Mtr4 is a part of the TRAMP complex (Trf4/5–Air1/2–Mtr4), which consists of Mtr4, a zinc-finger protein (Air1 or Air2), and a poly(A) polymerase (Trf4 or Trf5) (LaCava et al, 2005; Vanácová et al, 2005; Bühler et al, 2008). In *Saccharomyces cerevisiae*, the TRAMP complex is recruited to RNA co-transcriptionally by the RNA- and Pol II-binding protein Nrd1 and mediates degradation of CUTs, among other substrates (Thiebaut et al, 2006; Vasiljeva & Buratowski, 2006; Vasiljeva et al, 2008a, 2008b; Tudek et al, 2014). In contrast, the human and *S. pombe* TRAMP complexes appear to be more specialised in regulating rRNA processing (Houseley & Tollervey, 2009).

In fission yeast, Mtl1 interacts with the conserved YTH domain-containing protein Mmi1 and its associated proteins, the C3H1 type zinc-finger protein Red1, the proline-rich protein Iss10, and the enhancer of rudimentary homolog Erh1 (Sugiyama & Sugioka-Sugiyama, 2011; Lee et al, 2013; Yamashita et al, 2013; Egan et al, 2014; Zhou et al, 2015). Mmi1 is needed for degradation of a subset of mRNAs encoding for proteins involved in meiosis, cell cycle regulation, and RNA metabolism by the exosome complex and also targets specific ncRNAs. Within these transcripts, Mmi1 is co-transcriptionally recruited to UNAAAC sequence motifs, which can be enriched in regions known as "determinants of selective removal" (DSRs), leading to their rapid degradation by the exosome complex (Harigaya et al, 2006; Chen et al, 2011; Yamashita et al, 2012; Kilchert et al, 2015). The interaction of Mmi1 with Red1 and Mtl1 as a part of the MTREC/NURS (Mtl1-Red1 Core/Nuclear RNA Silencing) complex is important for mediating recruitment of the exosome to RNAs (Shichino et al, 2020). Mmi1 also acts in mRNA quality control and promotes the degradation of inefficiently spliced mRNAs and proper transcription termination of selected transcripts (Shah et al, 2014; Kilchert et al, 2015; Vo et al, 2019). In addition to Mmi1, Iss10, Erh1, and Red1, Mtl1 also co-purifies with other factors that have been functionally linked to exosome regulation: the zf-CCCH-type zinc-finger protein Red5, the poly(A)–binding protein Pab2, the RRM (RNA-Recognition-Motif) and PWI (Pro-Trp-Ile signature) domain–containing protein Rmn1, and the cap-binding proteins Cbc1, Cbc2, and Pir2 (Lee et al, 2013; Zhou et al, 2015). Nevertheless, the mechanisms by which these factors regulate substrate recognition and exosome targeting to substrate RNAs remain poorly understood.

In addition to the RNAs recognised by Mmi1, levels of multiple other transcripts increase in nuclear exosome mutants, suggesting that Mmi1-independent mechanisms contribute to their

recognition (Lee et al, 2013; Zhou et al, 2015). We therefore used a quantitative proteomics approach to identify RBPs that are involved in mediating exosome targeting to RNAs. We hypothesised that the association of exosome co-factors with RNA should increase upon stabilisation of their substrate RNAs in the exosome mutants. We compared the poly(A)+ transcriptomes and poly(A)+ RNA-bound proteomes of control cells and three different exosome mutants (*mtl1-1*, *rrp6Δ*, and *dis3-54*). Our data suggest that the nuclear exosome plays a more prominent role in controlling the fission yeast transcriptome than previously anticipated. In addition, analysis of the impact of mutations in different exosome subunits on the poly(A)+-bound proteome has identified proteins with increased RNA binding that could function as potential regulators of the exosome in fission yeast. We selected 10 RBPs with significantly altered RNA association and demonstrate that deletion of each of these proteins phenocopies at least one of various phenotypes typical of inactivated nuclear exosome, namely, suppression of transposon RNAs, telomeric silencing, and nuclear RNA retention. We focus on the uncharacterised zf-MYND (MYeloid, Nervy, and DEAF-1)-type zinc-finger protein Mub1, which is highly enriched on poly(A)+ RNA in the exosome mutants. Mub1 physically interacts with the exosome and its deletion leads to increased levels of a specific subset of exosome substrates, supporting its role in exosome regulation.

## Results

### Poly(A)+ RNA interactome capture (RIC) in exosome mutants

We had previously used an unbiased quantitative proteomics approach, RIC, to assess how mutation of various exosome components affects association of the entire complex with poly(A)+ RNA (Kilchert et al, 2020a). Specifically, we analysed: *rrp6Δ*, lacking exonuclease Rrp6; *dis3-54*, a Dis3 mutant containing an amino acid substitution (Pro509 to Leu509) within the RNB domain, which reduces its catalytic activity (Ohkura et al, 1988; Murakami et al, 2007); and *mtl1-1*, a mutant of the helicase Mtl1, which has mutations in the region surrounding the arch domain (Lee et al, 2013) (Fig 1A). All three mutants are defective in RNA turnover and accumulate known targets of the exosome complex (St-André et al, 2010; Lee et al, 2013; Shah et al, 2014). To gain further insights into the regulation of the exosome complex, we re-analyzed the RIC data to identify RBPs that show increased interactions with poly(A)+ RNA in these mutants. The underlying hypothesis behind this approach was that stabilisation of RNAs targeted by the exosome would facilitate the capture of proteins that are functionally linked to the exosome, which are likely to be enriched on these RNAs (Fig 1B). The three mutant strains were cultured alongside a wild-type (WT) control in the presence of 4-thiouracil (4sU) to facilitate RNA-protein cross-linking with 365 nm UV light. After UV crosslinking, poly(A)+ RNA was enriched by oligo-d(T) selection and poly(A)+ RNA-associated proteins were identified by mass spectrometry. The abundance of individual proteins in the whole cell extract (WCE) was used to normalise the RIC data as in Garcia-Moreno et al (2019) and Kilchert et al (2020a) (see the Materials and Methods section). The RIC/WCE

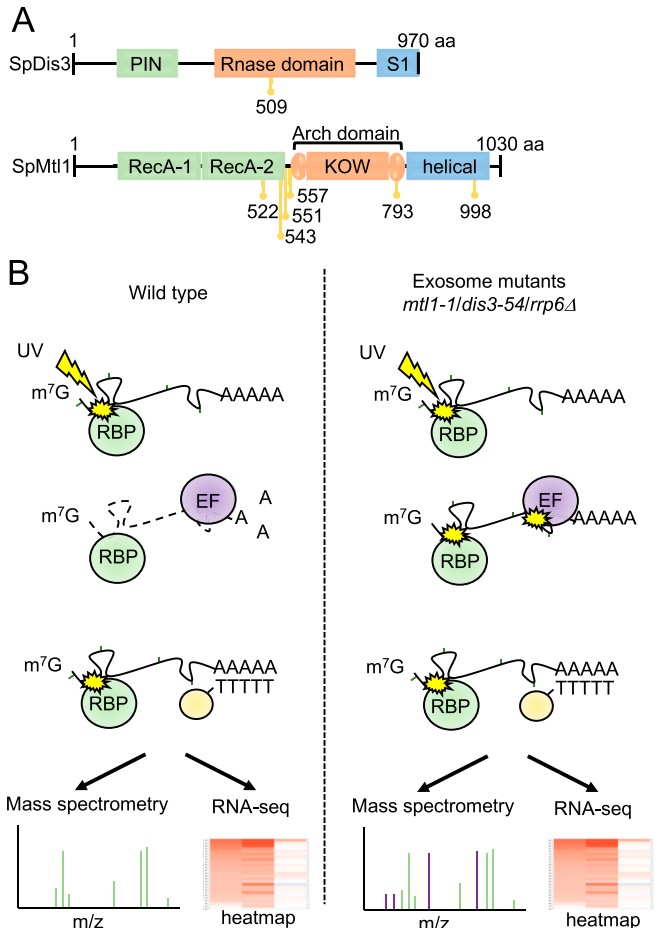

**Figure 1. Experimental design of the comparative poly(A)+ RNA interactome capture approach.**
**(A)** Schematic diagram of the domain organisation of *S. pombe* (Sp) Dis3 and Mtl1 with the position of the mutations in *dis3-54* (P509L) and *mtl1-1* (I522M, L543P, Y551H, L557P, D793G, and A998V) indicated in yellow. **(B)** Schematic diagram of the comparative poly(A)+ interactome capture approach. Cells are grown in the presence of 4-thiouracil (4sU) and exposed to UV (3 J/cm²) to allow protein-RNA crosslinking. Poly(A)+ RNA and associated proteins are enriched on oligo-d(T) beads and subjected to RNA sequencing and mass spectrometric analysis (RBP, RNA-binding proteins; EF, exosome co-factors involved in recognition of the exosome substrates).

ratio was used to determine the enrichment of each individual protein on poly(A)+ RNA in the mutants relative to the WT (Tables S1 and S2). In addition, RNA sequencing (RNA-seq) was carried out for the oligo-d(T) enriched samples to assess levels of individual poly(A)+ RNAs in WT cells and the exosome mutants.

### Exosome RNA targets are enriched in the poly(A)+ RIC samples of the exosome mutants

First, to confirm that exosome target RNAs were indeed overrepresented in the RIC samples of the different exosome mutants, we analysed their behaviour by RNA-seq. Consistent with the function of the exosome in degradation of ncRNAs (snRNAs, snoRNAs, and CUTs), levels of multiple nc transcripts were increased in all three exosome mutants (Fig 2A and Table S3). Compared to *rrp6Δ* and

*mtl1-1*, fewer RNAs in total increased in the *dis3-54* mutant (1,373, 1,718, and 623 for *mtl1-1*, *rrp6Δ*, and *dis3-54*, respectively; >1.5-fold, *P*-value < 0.05), possibly reflecting that Dis3 function is only partially inhibited under the condition tested. Approximately half of the transcripts with increased levels in the *dis3* mutant (54%) were also dependent on Mtl1, suggesting that these correspond to nuclear targets of the exosome. In agreement with previous reports, transcripts with increased levels in the mutants included many mRNAs (835, 1,078, and 317 for *mtl1-1*, *rrp6Δ*, and *dis3-54*, respectively; >1.5-fold, *P*-value < 0.05), confirming that the nuclear exosome regulates mRNA levels in addition to its well-described role in degradation of ncRNAs (Fig 2B–D) (Lee et al, 2013; Zhou et al, 2015; Atkinson et al, 2018). However, not all changes necessarily reflect increased mRNA half-lives that are directly related to impaired exosome activity but may be indirect consequences of altered exosome function. The majority of the transcripts that increased in *rrp6Δ* were also increased in *mtl1-1* (~80%), in agreement with the suggested functional connection between Mtl1 and Rrp6 (Lee et al, 2013; Shichino et al, 2020). Our data support a model where Mtl1 is functionally linked to both exosome-associated nucleases (Rrp6 and Dis3) – with Rrp6 being more dependent on Mtl1 than Dis3 – and confirmed that exosome target RNAs were enriched in the mutant RIC samples.

### Mutations in the exosome alter the RNA-bound proteome

To assess how inactivation of the different subunits of the exosome impacts the RNA-protein interaction profile, we next performed comparative analysis of RBPs differentially enriched in the interactomes of *mtl1-1*, *rrp6Δ* and *dis3-54* compared to WT. For each comparative interactome, proteins that were detected in at least two of three biological replicates were included in the analysis (see the Materials and Methods section), resulting in a quantitative data set for a total of 1,146 RBPs. 152, 180, and 83 RBPs were ≥2-fold enriched in *mtl1-1*, *rrp6Δ*, and *dis3-54* over WT, respectively (*P*-value < 0.1,Table S2). Consistent with the RNA-seq data, the overlap between RBPs with increased RNA association was substantially larger for *rrp6Δ* and *mtl1-1* than for *dis3-54* and *mtl1-1*, further supporting a functional link between Mtl1 and Rrp6 (Fig 2E). Conversely, poly(A)+ RNA association of 137, 95, and 86 RBPs decreased in *mtl1-1*, *rrp6Δ*, and *dis3-54*, respectively (*P*-value < 0.1, Fig 2F and Table S2). To assess whether RBPs that are differentially regulated in the three exosome mutants are linked to a specific biological process, we performed gene ontology (GO) analysis. The analysis revealed that nuclear RBPs are most noticeably affected in all three exosomes mutants, including *dis3-54* where both the nuclear and the cytoplasmic form of the exosome are compromised (Fig 2G and Table S4). This agrees with the prominent role of the exosome complex in nuclear RNA metabolism. In addition, all mutants exhibit alterations of RBPs involved in "mRNA metabolic process" (GO:0016071) and "ribosome biogenesis" (GO:0042254) (Table S4), suggesting a profound reorganisation of ribonucleoprotein complexes (RNPs) in the exosome mutants. In contrast, the GO term "cytoplasmic translation" (GO:0002181) was significantly overrepresented among the RBPs that contributed less to total RNA–protein interactions in the catalytic exosome mutants than in WT. Indeed, 22 of 95 RBPs that are underrepresented in *rrp6Δ* (*P*-value = 2.02 × 10⁻⁷) and 31

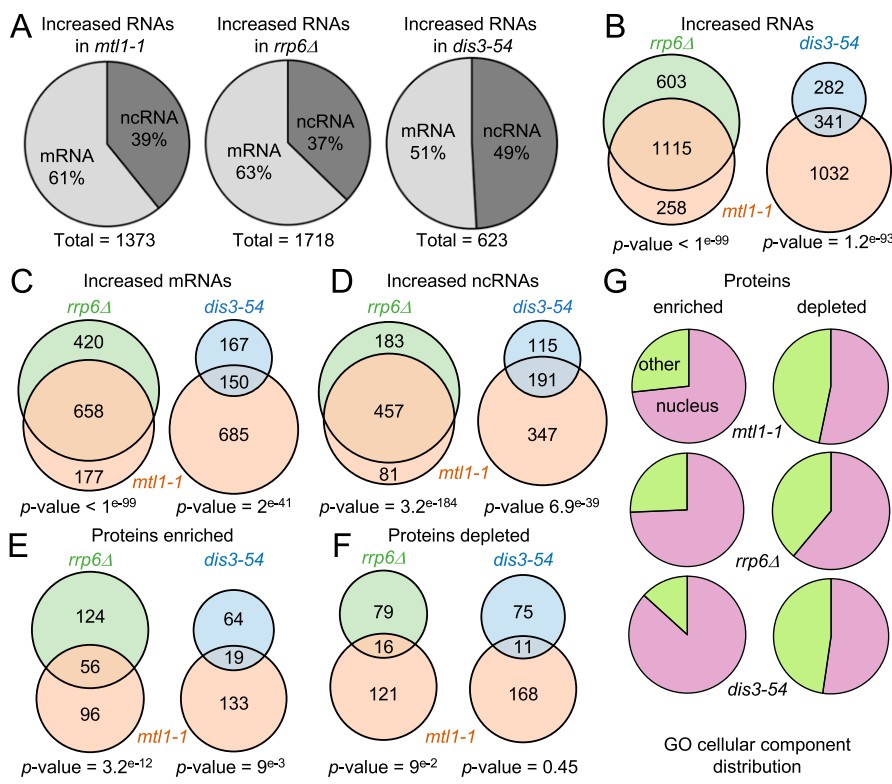

**Figure 2. Effect of exosome mutations on the poly(A)+ RNA transcriptome and protein interactome.**
**(A)** mRNAs and ncRNAs up-regulated (>1.5-fold, *P*-value < 0.05) in *mtl1-1*, *rrp6Δ*, and *dis3-54* mutants compared to WT. **(B)** Venn diagram showing overlap between RNAs with increased levels (>1.5-fold, *P*-value < 0.05) in *mtl1-1*, *rrp6Δ*, and *dis3-54*. The *P*-values indicate the probabilities that the observed overlaps occurred by chance (6,949 genes analysed). **(C)** Venn diagram showing overlap between mRNAs with increased levels in *mtl1-1*, *rrp6Δ*, and *dis3-54* mutants compared to WT (>1.5-fold, *P*-value < 0.05). The *P*-values indicate the probabilities that the observed overlap occurred by chance (5,153 mRNAs analysed). **(D)** Venn diagram showing overlap between ncRNAs with increased levels in *mtl1-1*, *rrp6Δ*, and *dis3-54* mutants compared to WT (>1.5-fold, *P*-value < 0.05). The *P*-values indicate the probabilities that the observed overlap occurred by chance (1,796 ncRNAs analysed). **(E)** Venn diagram showing proteins that are enriched in poly(A)+ RNA pull-downs of *mtl1-1* and either *rrp6Δ* or *dis3-54* relative to WT. The *P*-values indicate the probabilities that the observed overlap occurred by chance (1,146 proteins analysed). **(F)** Venn diagram showing proteins depleted from the poly(A)+ pull-down in *mtl1-1* and either *rrp6Δ* or *dis3-54* relative to WT. The *P*-values indicate the probabilities that the observed overlap occurred by chance (1,146 proteins analysed). **(G)** RNA association of nuclear proteins (GO:0005575) is strongly affected in the exosome mutants.

of 86 RBPs that are underrepresented in *dis3-54* (*P*-value = $7.16 \times 10^{-17}$) are annotated with "cytoplasmic translation." Altogether, RIC analyses of exosome mutants demonstrate significant changes in protein–RNA interactions in exosome-deficient cells, highlighting the important role of the exosome in nuclear RNP metabolism.

## Exosome mutants show different patterns of poly(A)+ RNA accumulation in the nucleus

The strong increase in the association of poly(A)+ RNA with RBPs in the nucleus combined with the lower association with proteins involved in cytoplasmic translation may reflect a change in sub-cellular localisation of poly(A)+ RNA in these strains. Indeed, it was previously reported that poly(A)+ RNA is retained in the nucleus in *rrp6* mutants (Paul & Montpetit, 2016; Fan et al, 2017; Silla et al, 2018).

To assess whether all three exosome mutants exhibit increased levels of poly(A)+ RNA in the nucleus, we performed FISH with oligo(dT) probes. Poly(A)+ RNA accumulated inside the nucleus in all three mutants (Fig 3A). Quantification of the FISH signal confirmed a phenotype of nuclear RNA retention manifested by a higher nuclear/cytoplasmic ratio compared to WT, with the strongest effect observed for *rrp6Δ* (Fig 3B and Table S5). However, careful observation revealed different patterns of poly(A)+ RNA accumulation in the three mutants (Fig 3A): In *dis3-54*, a diffuse nuclear signal is observed. In contrast, distinct poly(A)+ RNA foci are detected inside the nuclei of *mtl1-1* cells. Last, *rrp6Δ* cells show a mixed phenotype with foci present in addition to a strong diffuse nuclear poly(A)+ FISH signal. We

had previously reported that RNA association with the exosome complex is increased in *dis3-54* (reflecting its inability to efficiently degrade RNA substrates), whereas RNA association with the exosome complex is strongly reduced in *mtl11*, suggesting a disruption of substrate recruitment as well as decay (Kilchert et al, 2020a). This blockage at different stages of the RNA degradation process may be responsible for the different be-haviour of the retained transcripts. An accumulation of exo-some target RNAs in nuclear foci has been observed upon inactivation of hMTR4 or its co-factors in human cells (Meola et al, 2016; Fan et al, 2018).

## Comparative RIC identifies known exosome co-factors

To validate our hypothesis that exosome co-factors are indeed differentially enriched in the comparative interactome capture experiment, we first analysed how RNA association of Mmi1 is af-fected in the exosome mutants. As expected, the association of Mmi1 with poly(A)+ RNA increased two and fourfold in *mtl1-1* and *rrp6Δ* RIC, respectively, compared with WT cells (*P*-value = 0.039 and *P*-value = 0.001); in *dis3-54*, the change was not significant (>1.5-fold, *P*-value = 0.768) (Fig 3C). The increased RNA association of Mmi1 correlated with an increased abun-dance of Mmi1 RNA substrates in the exosome mutants. Well-characterised meiotic RNA targets of Mmi1 were most strongly enriched in *rrp6Δ* cells, significantly enriched in the *mtl1-1* mutant, and only mildly affected in *dis3-54* (Fig 3D). Moreover, the relative poly(A)+ RNA association of other known exosome co-factors also increased in the exosome mutants. For example,

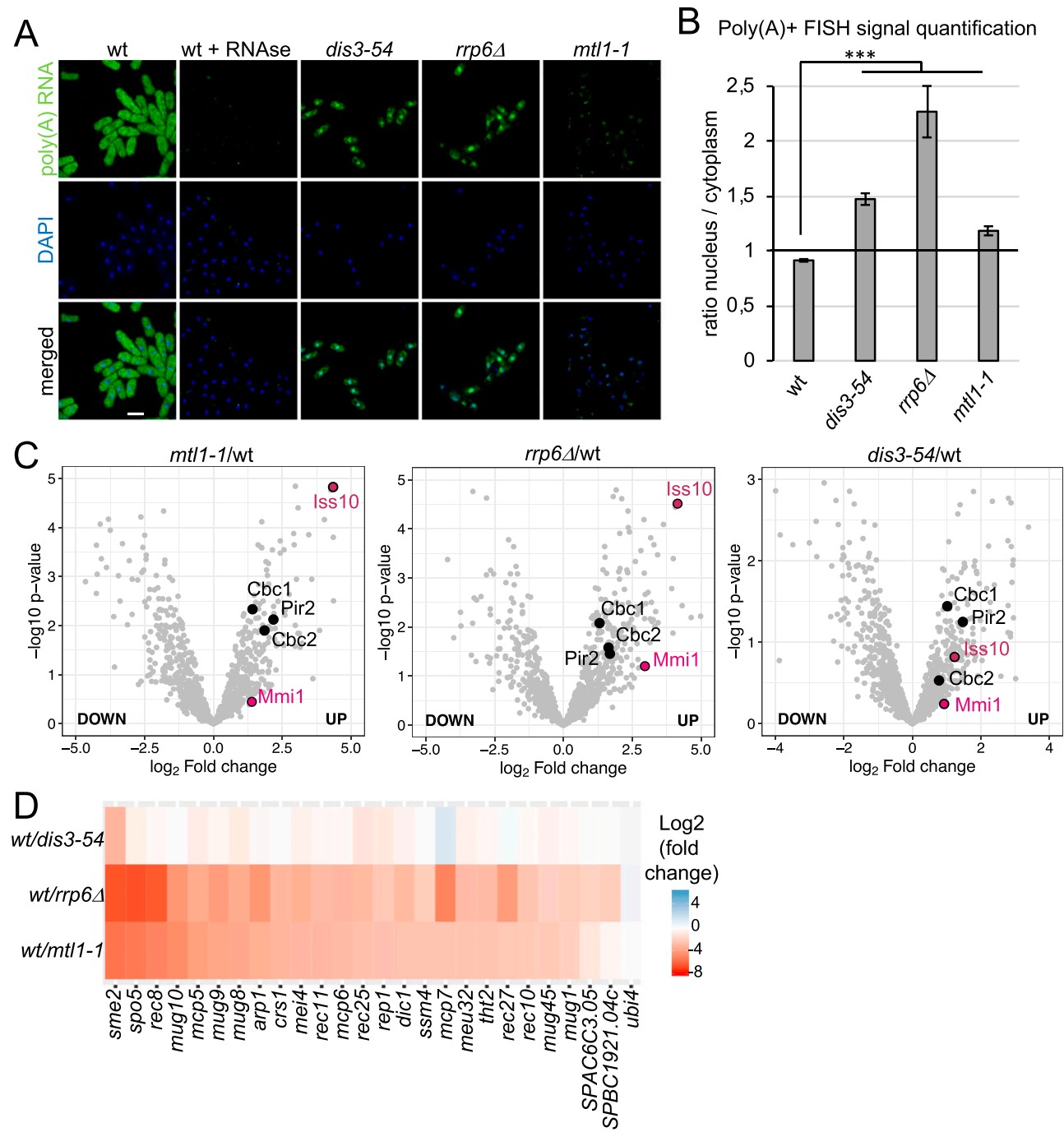

**Figure 3. Poly(A)+ RNA accumulates in the nucleus of exosome mutants and RBPs with functional links to the exosome are enriched on poly(A)+ RNA.**
**(A)** Poly(A)+ RNA FISH analysis of *rrp6Δ*, *dis3-54*, and *mtl1-1* cells. Nuclear co-staining with DAPI is shown in blue. Poly(A)+ RNA is visualised in green. Scale bar = 10 μm.
**(B)** Quantification of the poly(A)+ RNA FISH experiments shown in Fig 3A (n = 30–60 cells, ***P-value < 0.0001, un-paired *t* test). **(C)** Volcano plots showing enrichment of Mmi1, Iss10, Cbc1, Cbc2, and Pir2 (*S. pombe* ARS2) on poly(A)+ RNA in the three exosome mutants. In the volcano plot, *P*-values (−log10, moderated *t* test) are plotted against the ratio of log₂-fold changes in mass spectrometry (MS) intensities for the whole-cell extract-normalised proteomes of mutant versus WT cells recovered from the oligo-d(T) pull-downs of UV-crosslinked samples (3 J/cm²). In all panels, individual proteins are depicted as a single dot. **(D)** Heat map analysis of poly(A)+ RNA-seq showing differential expression of Mmi1 regulon RNAs in exosome mutants compared with WT.

RNA association of Iss10, an MTREC component, increased between 2-fold and more than 10-fold in *dis3-54*, *mtl1-1*, and *rrp6Δ* over WT (*P*-value = 0.15, *P*-value = 3.26 × 10⁻⁵ and *P*-value = 1.73 × 10⁻⁵). Similarly, RNA association of the nuclear cap-binding complex Cbc1,

Cbc2 and Pir2 increased (>4-fold) in all three mutants (Table S1). Together, this illustrates the capacity of comparative RIC to discover RBPs with bona fide roles in exosome targeting to substrate RNAs.

## Identification of RBPs with functional links to the exosome

To uncover cellular RBPs with novel roles in exosome regulation, we selected 10 proteins without well-described functions in RNA metabolism that have a classical RNA-binding domain (Lunde et al, 2007) and whose RNA association was increased (>2-fold) in the exosome mutants (Figs 4A and S1A). We generated genomic deletions of the candidate RBPs to test whether these recapitulate phenotypes associated with compromised exosome function. First, we assessed whether nuclear retention of poly(A)+ RNA is observed in any of the deletion strains using RNA FISH (Figs 4B and S1B). In 8 of 10 deletion strains, the nuclear/cytoplasmic ratio of the poly(A)+ RNA FISH signal was significantly increased relative to WT (Fig 4C and Table S6), with the strongest effect observed for *SPCC126.11c*Δ (ratio of poly(A)+ signal ~1.5). In all cases, the pattern of RNA accumulation in the nucleus resembled the diffuse staining of *rrp6*Δ and *dis3-54* rather than the pronounced punctate staining characteristic of the *mtl1-1* mutant (compare Fig 3A). No nuclear RNA retention was observed upon deletion of *mub1* or *SPABC18H10.09*.

The exosome complex is also required for robust heterochromatic gene silencing in *S. pombe*, resulting in the accumulation of heterogeneous RNAs produced from telomeric and centromeric regions of the genome in exosome mutants (Bühler et al, 2007; Lee et al, 2013). To test whether the candidate RBPs contribute to the repression of heterochromatic transcripts, the deletion mutants were crossed into a strain background with a *ura4+* reporter inserted within the transcriptionally silent telomeric region of chromosome I. We then monitored the ability of the reporter strains to grow on –URA plates and plates containing 5-fluoroorotic acid (5-FOA), a compound that is toxic when a functional version of the *ura4* gene is expressed. The deletion of either *SPCC126.11c*, *srp40*, or *SPAC222.18* led to moderate growth on –URA plates and attenuated growth on 5-FOA compared with WT (Figs 4D and S1C), suggesting that these RBPs may function in heterochromatin formation or maintenance.

To assess whether any of the 10 candidate RBPs contributes to regulating levels of exosome substrate RNAs, we determined the steady-state levels of the RNA produced from the Tf2 retrotransposable element (*SPAC9.04*). Tf2-1 is up-regulated in *rrp6*Δ but independent of Mmi1 (Fig 4E, lanes 1–5). Only in *mub1*Δ cells was a pronounced increase in tf2-1 RNA levels observed, suggesting that, similar to Rrp6, Mub1 may contribute to the suppression of RNAs derived from retrotransposable elements (Fig 4E).

## Mub1 controls levels of stress-induced mRNAs targeted by the exosome

Because deletion of *mub1* led to increased levels of known exosome targets such as tf2-1 RNA, we investigated the involvement of the protein in RNA metabolism in more detail. Fission yeast Mub1 is the homolog of *S. cerevisiae* multi-budding 1 and contains an Armadillo-type domain and a potential nucleic acid-binding region (zf-MYND domain). To assess the contribution of Mub1 to regulation of RNA levels, we compared the transcriptomes of WT and *mub1*Δ cells by RNA-seq normalised to *S. cerevisiae* spike-in (see the Materials and Methods section). The levels of 248 transcripts (162 mRNAs and 86 ncRNAs) increased more than 1.5-fold (*P*-value < 0.05) in

*mub1*Δ cells (Fig 5A and Tables S7 and S8). GO term analysis revealed a significant enrichment of "Core Environmental Stress Response induced (CESR)" genes (46.25% (74/160), *P*-value = 1.55246 × 10$^{-28}$, AnGeLi tool [Bitton et al, 2015]). 144 transcripts (92 mRNAs and 52 ncRNAs) affected by Mub1 deletion were also up-regulated in *mtl1-1* (Figure 5A and B), in support of a possible functional link between Mtl1 and Mub1. The GO term "Core Environmental Stress Response induced" was strongly overrepresented among mRNAs that were upregulated in both mutant strains (64.44% (58/90), *P*-value = 3.32315 × 10$^{-32}$, AnGeLi tool [Bitton et al, 2015]), potentially indicating a role for Mub1 and Mtl1 in repressing the stress response under non-stress conditions. To validate the RNA-seq data, increased steady-state levels of two representative CESR mRNAs (Chen et al, 2003), gst2 and SPCC663.08c, in *mub1*Δ and *mtl1-1* mutants were confirmed by Northern blot analyses (Fig 5B). No additive effect was observed in the double mutant *mtl1-1 mub1*Δ compared with the single mutants, which is consistent with Mtl1 and Mub1 acting in the same pathway (Fig 5B).

## The MYND-type zinc finger domain is required for regulation of RNA levels

Our observation that Mub1 was enriched in the poly(A)+ RNA interactome suggested that Mub1 is an RBP. We hypothesised that the zf-MYND domain (471-528) could mediate an interaction with RNA. Multiple other proteins that associate with the exosome contain zinc-finger domains, for example Red1, Red5, and Air1, a component of the TRAMP complex (Keller et al, 2010; Sugiyama & Sugioka-Sugiyama, 2011; Sugiyama et al, 2013). For Red1 and Red5, the zinc-finger domains were shown to be important for RNA degradation by the exosome as mutations introduced in these domains lead to RNA stabilisation (Sugiyama & Sugioka-Sugiyama, 2011; Sugiyama et al, 2013). To test whether the zf-MYND domain is required for Mub1 function, we deleted the region between amino acids 471 and 528 (Fig 5C). Deletion of the zf-MYND domain led to the expected change in the size of the protein, which we assessed by visualising FLAG-tagged protein by Western blot (Fig 5D). Addition of a triple FLAG tag to full-length Mub1 did not have any effect on cellular protein levels or cell growth, nor did it induce the spheroid cell shape typical of a *mub1* deletion (Hayles et al, 2013) (Figs 5E and S2A and B). The protein level of Mub1Δ471-528-3xFLAG (Mub1-ΔZ-3xFLAG) was comparable to Mub1-3xFLAG, suggesting that deletion of the zf-MYND domain did not affect the stability of the truncated protein (Fig 5D). To test the importance of the zinc-finger domain for regulation of mRNA levels, we assessed levels of gst2 and SPCC663.08c mRNAs in the *mub1-ΔZ-3xFLAG* mutant by Northern blot. Similar to the phenotype observed for *mub1*Δ, expression of both mRNAs increased in the *mub1-ΔZ-3xFLAG* mutant compared with WT cells (Fig 5F), suggesting that the zf-MYND domain is essential for the function of Mub1 in RNA metabolism. At present, we do not know to what extent disruptions of either protein–RNA or protein–protein interactions contribute to the mutant phenotype. Cells expressing the truncated Mub1-ΔZ-3xFLAG display the characteristic rounded shape of *mub1*Δ cells (Fig S2A and data not shown), a phenotype that is not observed for exosome mutants, suggesting that additional, exosome-independent functions of the protein might also be affected in this mutant.

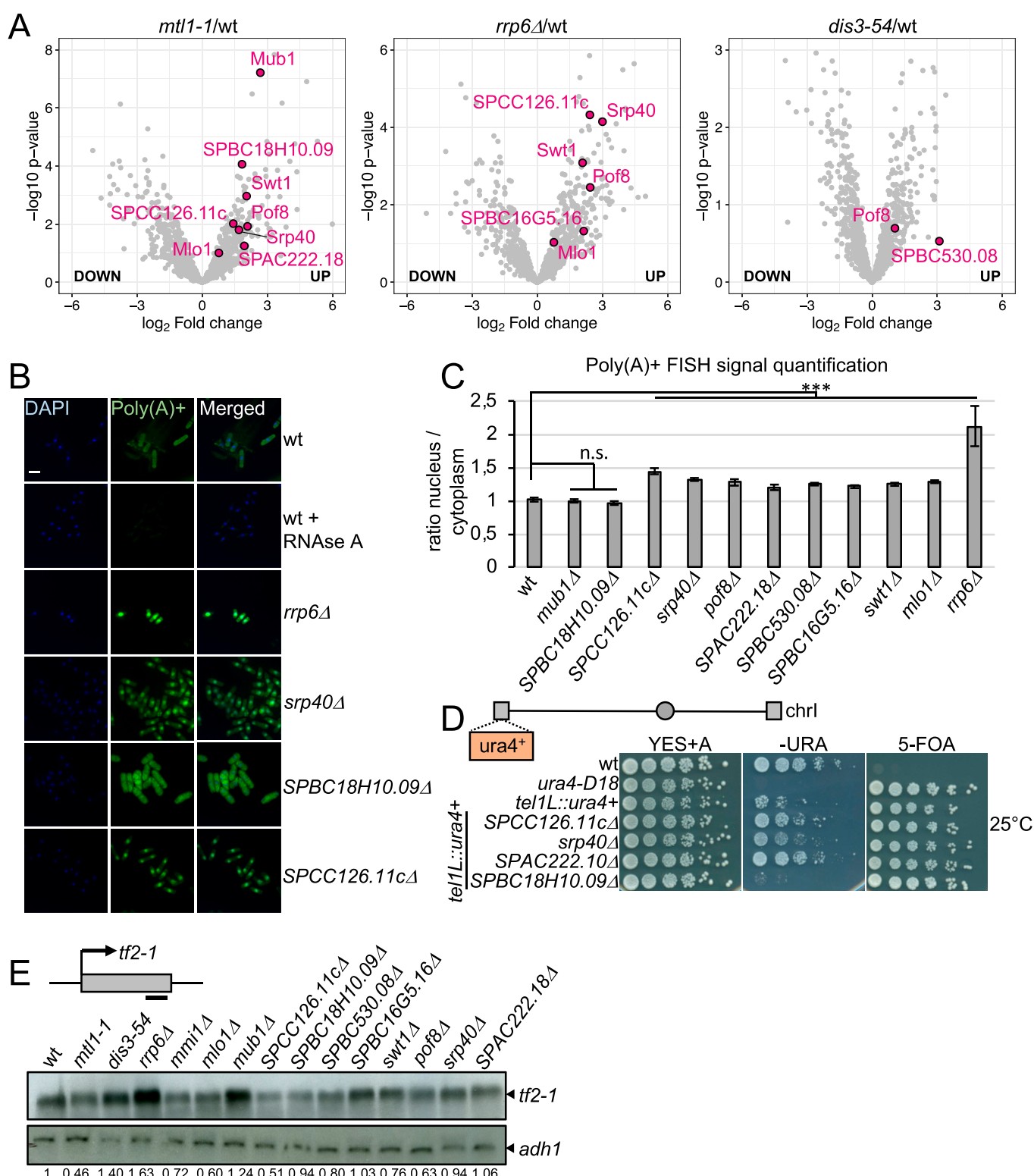

**Figure 4. Identification of RBPs with functional links to the exosome.**
**(A)** Volcano plot showing the relative enrichment of the candidate proteins on poly(A)+ RNA in *mtl1-1*, *rrp6Δ*, and *dis3-54* as in 3C. **(B)** Poly(A)+ RNA FISH analysis of the indicated strains. Nuclear co-staining with DAPI is shown in blue. Poly(A)+ RNA is visualised in green. Scale bar = 10 μm. **(C)** Quantification of the poly(A)+ RNA FISH experiments shown in Figs 4B and S1B (n = 25–60 cells, ***P-value < 0.0001, n.s., not significant, un-paired *t* test). **(D)** Serial dilutions of the indicated strains spotted on complete medium supplemented with adenine (YES+A), medium lacking uracil (−URA) or containing 5-FOA (5-FOA) and grown at 25°C. A schematic representation of Chromosome I (Chr I) with centromeric and telomeric regions (grey circle and grey rectangles, respectively) is included. The position of the *ura4+* reporter in the left arm of

### Disruption of Mub1 function leads to sensitivity to heat stress

We had observed that genes involved in the core environmental stress response—many of which are induced during heat shock—are dysregulated in *mub1Δ* and *mtl1-1* already under non-stressed conditions (see above). To test whether the dysregulation of stress-responsive transcripts extended to stress conditions, we performed Northern blot for hsp16 mRNA, which encodes a heat shock protein (Hirose et al, 2005). Under non-stressed conditions, no detectable RNA is produced from the *hsp16* locus in WT cells; however, several bands can be detected in *mub1Δ* and *mtl1-1* (Fig S2C and D, lanes 1–4). The lower band corresponds to the mature mRNA in size, whereas the upper bands may represent isoforms with longer 5′ or 3′UTRs. As expected, levels of hsp16 mRNA sharply increased in WT cells after induction of the heat shock response (4 h at 37°C). However, heat shock-dependent induction of hsp16 mRNA was amplified in *mub1Δ*, *mtl1-1*, and the double mutant (Fig S2D, compare lanes 6–8 to lane 5). Please note that the hsp16 isoforms that are stabilised in the mutants under non-stressed conditions are absent at 37°C, potentially reflecting a higher complexity in *hsp16* regulation, which needs to be further characterised. When we assayed the comparative fitness of the different strains under conditions of heat stress, we observed that growth at 37°C is impaired in *mub1Δ* and *mub1-ΔZ-3xFLAG* (Fig 5E). We consider it plausible that the dysregulation of expression of stress-responsive genes contributes to the increased sensitivity of *mub1* mutant strains to high temperatures, although we cannot exclude that Mub1 might also promote cellular adaptation to suboptimal temperatures through other mechanisms.

### Mub1 regulates levels of stress-responsive transcripts at a post-transcriptional level

We observed a strong de-repression of many CESR genes, including *gst2*, *SPCC663.08c*, and *hsp16*, in *mub1Δ* and *mtl1-1* mutants. To test whether mutation of *mub1* triggers a transcriptional induction of the CESR under normal growth conditions, we assessed Pol II occupancy at these genes by chromatin immunoprecipitation (Pol II ChIP). We observed no significant change in Pol II occupancy in *mub1Δ* and *mub1-ΔZ-3xFLAG* compared with WT cells (Fig 6A and Table S9). These results strongly suggest that Mub1 contributes to regulation of stress-responsive transcripts at the post-transcriptional level.

To assess whether Mub1 and the exosome complex physically interact, we performed co-immunoprecipitation experiments. Mub1 co-purified with the Rrp6 subunit of the exosome, supporting a physical link between Mub1 and the nuclear exosome (Fig 6B). The reverse co-immunoprecipitation yielded the same result (Fig 6C). Importantly, the interaction between Rrp6 and Mub1 is no longer detected upon RNAse A treatment (Fig 6C). These data support a connection between Mub1 and the exosome complex that is mediated by RNA and is compatible with Mub1 acting as a co-factor of the complex.

## Discussion

Taken together, our comparative RIC analysis identified several proteins that may function as potential exosome regulators, including Mub1, SPCC126.11c, Srp40, or SPAC222.18, whose deletion phenocopies key aspects of exosome dysfunction. Based on the correlated phenotypes, our data suggest a relationship between these factors and the nuclear RNA exosome. However, further work is needed to fully explain and understand how these factors might be functionally connected. Interestingly, SPCC126.11c and Srp40, which we did not further characterize, were previously reported to co-purify with the exosome, suggesting a direct functional link between these proteins and the exosome (Egan et al, 2014; Telekawa et al, 2018). SPCC126.11c has some similarity to ALYREF and CHTOP, components of the human TRanscription and EXport (TREX) complex that also co-purify with hMTR4 in human cell lines (Gaudet et al, 2011; Andersen et al, 2013). Moreover, a recent study reports that SRSF3, the human orthologue of SPAC222.18, is involved in exosome recruitment via the Nuclear Exosome Targeting (NEXT) complex, raising the possibility that the role of the serine-rich protein SPAC222.18 is conserved in *S. pombe* (Mure et al, 2018). In particular, our data highlight a role for Mub1 and the exosome complex in suppressing the heat shock response, potentially through rapid turnover of stress-responsive mRNAs under non-stressed conditions. A reverse mechanism was recently described in budding yeast, where the Rrp6 component of the exosome was found to be required for the full induction of cell wall integrity (CWI) genes, thereby promoting cell survival during heat shock (Wang et al, 2020). Similarly, the exosome was shown to be recruited to heat shock genes in heat-stressed *Drosophila* Kc cells (Andrulis et al, 2002). This suggests that a function of the exosome in modulating environmental adaptation is conserved and can act on multiple levels. Further works will assess by which mechanism Mub1 contributes to this regulation.

## Materials and Methods

### Yeast strains and manipulations

General fission yeast protocols and media are described in Moreno et al (1991). All strains are listed in Table S10. Cells were grown in YES medium at 30°C unless stated otherwise. Gene deletions and epitope tagging were carried out by homologous recombination (Bähler et al, 1998). All oligos are listed in Table S11. Protein extracts and Western blotting were performed as described in Feytout et al (2011).

---

the telomere is indicated. **(E)** Analysis of tf2-1 mRNA levels by Northern blot in the indicated strains. Band corresponding to tf2-1 is indicated with an arrow. Schematic shows location of the probe (black bar). Adh1 mRNA is included as a loading control. Numbers indicate the relative abundance of tf2-1 RNA normalised to adh1 (technical replicate n = 2).

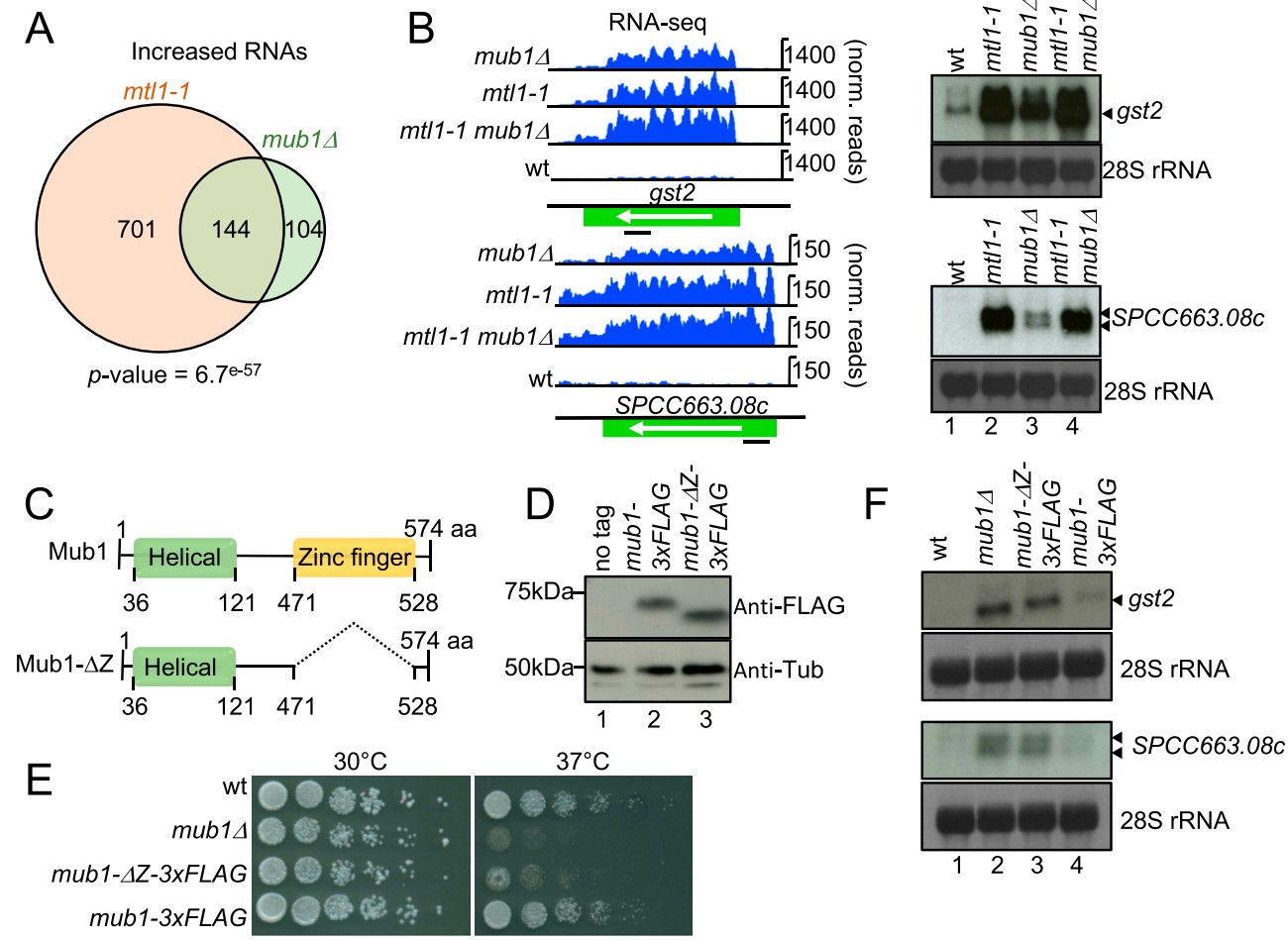

**Figure 5. Mub1 protein functions in maintaining levels of exosome RNA targets.**
**(A)** Venn diagram showing overlap between RNAs that increase in levels in *mtl1-1* and *mub1Δ* (>1.5-fold, *P*-value < 0.05). **(B)** Genome browser snapshots showing RNA-seq data and Northern blot analysis for two representative transcripts (gst2 and spcc663.08c) up-regulated in *mtl1-1* and *mub1Δ* compared with WT. Bands corresponding to gst2 and spcc663.08c are indicated with arrows. Positions of the probes are indicated with black bars. 28S ribosomal RNAs (rRNA) stained with methylene blue are included as loading control (biological replicate n = 3). **(C)** Graphical representation of Mub1 and Mub1-ΔZ. **(D)** Western blot against FLAG to assess levels of endogenously expressed Mub1-3xFLAG and Mub1-ΔZ-3xFLAG proteins (biological replicates n = 2). **(E)** Serial dilutions of the indicated strains spotted on complete medium (YES+A) at the indicated temperatures (biological replicates n = 2). **(B, F)** Northern blot analysis of SPCC663.08c and gst2 mRNAs as in (B), (biological replicates n = 2).

## Co-immunoprecipitation

To prepare WCEs, cells were collected, rinsed in ice-cold PBS with1 mM PMSF, and frozen on dry ice. Lysis was performed in 125 µl ice-cold lysis buffer (50 mM Tris, pH 7.5, 150 mM NaCl, 1 mM MgCl₂, 0.5% Triton X-100, 1 mM DTT, and 10% glycerol) with inhibitors (complete EDTA free protease inhibitor, 40694200; Roche, and 1 mM PMSF) using a bead beater (Cryoprep). After addition of 275 µl ice-cold lysis buffer with protease inhibitors, the extracts were clarified by two successive rounds of centrifugation. Samples were incubated for 1 h at 4°C with anti-MYC magnetic beads (TA150044; Origene). Beads were washed five times with lysis buffer without inhibitors. Proteins were eluted with hot Laemmli buffer at 95°C for 5 min. For reverse co-immunoprecipitation, lysis was performed in ice-cold buffer (50 mM Hepes, pH 7.6, 75 mM KCl, 1 mM MgCl₂, 1 mM EGTA, 1 mM DTT, 0.1% Triton X-100, 10 mM Sodium butyrate, and 10% glycerol) with inhibitors (1 mM PMSF, 1 mM NaVa, 10 mM β-glycerophosphate, and 1X

protease inhibitor [P8215; Sigma-Aldrich]). Samples for each strain were distributed to two tubes. One was treated with 2.5 µg of RNase A (R6513; Sigma-Aldrich) for 20 min on ice. Samples were immunoprecipitated for 1 h at 4°C with anti-FLAG antibody (F1804; Sigma-Aldrich) coupled to µMACS protein G (130-071-101; Miltenyi Biotec).

## Northern blotting

Northern blot experiments were essentially performed as described (Vasiljeva & Buratowski, 2006). RNA was prepared as described in Kilchert et al (2015). 8 µg of RNA were resolved on a 1.2% agarose gel containing 6.7% formaldehyde in MOPS buffer. After capillary transfer in 10× SSC onto a Hybond N+ membrane (GE Healthcare), RNA was UV-crosslinked and stained with methylene blue to visualise ribosomal RNAs. Gene-specific probes were generated by random priming in the presence of ATP [α³²P] using the Prime-It II Random Primer Labeling Kit (300385; Agilent) using

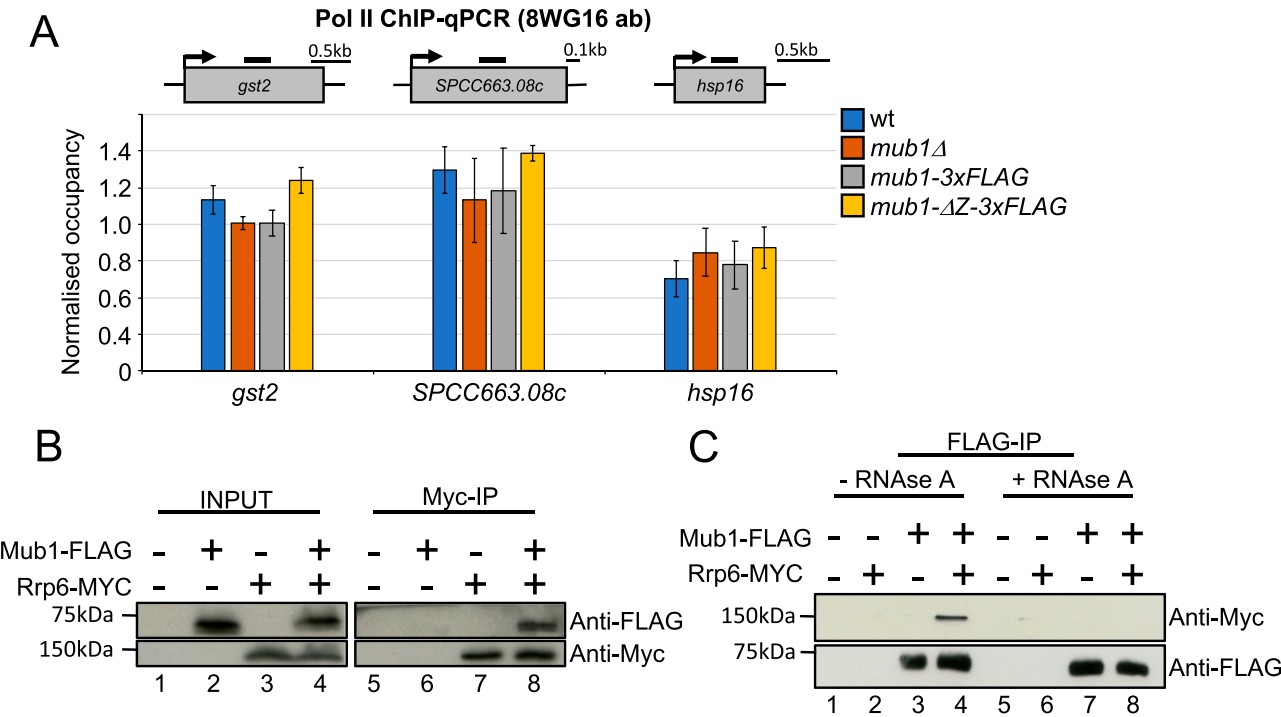

**Figure 6. Mub1 regulates stress-dependent genes post-transcriptionally and physically associates with the exosome complex.**
**(A)** ChIP-qPCR analysis of RNA polymerase II (antibody 8WG16) levels at *gst2*, *SPCC663.08c* and *hsp16* in WT, *mub1Δ*, *mub1-3XFLAG*, and *mub1-ΔZ-3XFLAG*. The diagram shows the organisation of the genes and the positions of the primer pairs used for qPCR (black bars). ChIP-qPCR quantification shown as the ratio of IP over input relative to a control gene (*fbp1*). Error bars represent the standard error of the mean (SEM) (biological replicates n = 3). **(B)** Co-immunoprecipitation of Rrp6-Myc with Mub1-3xFLAG (biological replicates n = 2). **(C)** Co-immunoprecipitation of Mub1-3xFLAG with Rrp6-Myc from samples that were or were not treated with RNase A, as indicated (biological replicates n = 2).

PCR-generated DNA templates produced from genomic DNA isolated from a wild type *S. pombe* strain (YP71) with the oligonucleotides listed in Table S11. Probes were added to the membrane and hybridised at 42°C overnight. After repeated washes in 2× SSC, 0.1% SDS, blots were exposed to Amersham Hyperfilm MP (28-9068-44; GE Healthcare).

### Poly(A)+ RIC

*S. pombe* poly(A)+ RIC was performed as described (Kilchert et al, 2020a, 2020b). Briefly, two sets of triplicate experiments (wild-type 1 (WT1) + *mtl1-1* = First eXperiment (FX); WT2 + *rrp6Δ* + *dis3-54* = Second eXperiment (SX)) were performed. *S. pombe* cells were grown at 30°C in Edinburgh minimal media supplemented with glutamic acid (EMMG) with limited amounts of uracil (10 mg/l) and labelled with 4-thiouracil (1 mg/l) for 4 h 30 min. Cells were harvested by filtration, snap-frozen in liquid nitrogen after UV-crosslinking at 3 J/cm$^2$ in 50 ml PBS and lysed by grinding in liquid nitrogen. The grindate was resuspended in oligo-d(T) lysis buffer (20 mM Tris–HCl, pH 7.5, 500 mM LiCl, 0.5% Lithium Dodecyl Sulfate [LiDS], 1 mM EDTA; 5 mM DTT, protease inhibitor cocktail IV (fungal) 1:10,000; and 5 mM DTT) and cleared by centrifugation. 1/48[th] of the total volume of WCE was used for proteomics analysis, the rest was subjected to pull-down with oligo-d(T)x25 magnetics beads (NEB-S1419S), 1 ml of slurry per 1 liter of cell culture during 1 h at 4°C. Immobilised oligo-d(T)x25 magnetics beads were washed two times with oligo-d(T)

wash buffer 1 (20 mM Tris–HCl, pH 7.5, 500 mM LiCl, 0.1% LiDS, 1 mM EDTA, and 5 mM DTT) at 4°C, two times with oligo-d(T) wash buffer 2 (20 mM Tris–HCl pH 7.5, 500 mM LiCl, and 1 mM EDTA) at room temperature and two times with oligo-d(T) low salt buffer (20 mM Tris–HCl, pH 7.5, 200 mM LiCl, and 1 mM EDTA) at room temperature. RNA–protein complexes were eluted from beads with oligo-d(T) elution buffer (20 mM Tris–HCl, pH 7.5, and 1 mM EDTA), 330 µl/l of culture for 10 min at 55°C. 1/33 of the oligo-d(T) pull-down total volume was used for RNA sequencing analysis. The rest was subjected to RNase-A and RNase T1 treatment and subjected to mass spectrometry analysis as described in Kilchert et al (2020a).

### Statistical data analysis

Statistical analysis was performed essentially as described (Kilchert et al, 2020a) with the following modifications. To be considered for analysis, proteins were required to be present in at least one of the interactomes with two non-zero values. Raw intensities were log$_2$ transformed, normalised to the same median, and analysis was followed by the imputation of missing values using a minimal value approach (MinDet–where each sample is considered independently). Data manipulations, principal component analysis, and Pearson correlation plots were performed with the DEP package implemented in R (Zhang et al, 2018). Median-normalised data values were used to estimate the log-fold changes between exosome mutants and WT cells, which were further normalised to the

whole-cell extracts (WCE-normalisation). To minimise batch effects, control experiments (WT cells) were performed twice in triplicates alongside each set of exosome mutants (First triplicate = FX = WT1 + *mtl1-1*; Second triplicate = SX = WT2 + *rrp6Δ* + *dis3-54*). To test the changes between whole-cell extract-normalised (WCE-normalised) proteomes of mutants and WT cells, we used modified scripts from the DEP package. Briefly, this software takes advantage of the Limma package that calculates moderated t-statistics on a linear model fit to the expression data (Zhang et al, 2018). It allows defining custom contrasts (e.g., comparing difference of differences—as in the case of the WCE-normalised intensities). Proteins with a $\log_2$ (WCE-normalised RIC of exosome mutant/WCE-normalised RIC of WT) > 1 were considered to be specifically enriched in exosomes mutants. All other analyses were performed with custom scripts or ones modified from the DEP package. *S. pombe* GO term annotations and information on individual proteins were retrieved from PomBase (Lock et al, 2019).

### Poly(A)+ RNA fluorescence *in situ* hybridization (FISH)

Poly(A)+ RNA FISH was carried out as described (Heinrich et al, 2017; Trcek et al, 2017), using an oligo-d(T)x20-alexa488 (7206906; Invitrogen) DNA probe. Briefly, $5 \times 10^7$–$1 \times 10^8$ cells were used per hybridization reaction. Cells from an asynchronously growing culture were fixed by the addition of paraformaldehyde to the culture to a final concentration of 4%. The cell pellet was washed with 1 ml of buffer B (1.2 M sorbitol and 100 mM $KH_2PO_4$ at pH 7.5, 4°C), immediately resuspended in 1 ml of spheroplast buffer (1.2 M sorbitol, 100 mM $KH_2PO_4$ at pH 7.5, 20 mM vanadyl ribonuclease complex, and 20 $\mu$M $\beta$-mercaptoethanol) with 1% 100T zymolyase (083209-CF; MP Biomedicals) and incubated for 60 min to digest the cell wall. The reaction was stopped by washing with 1 ml of cold buffer B. Cells were incubated for 20 min in 0.01% Triton X-100/1X PBS and washed with 10% formamide/2× SSC at room temperature. Before hybridization, 50 ng of the oligo-d(T) probe was mixed with 2 $\mu$l of a 1:1 mixture between yeast transfer RNA (10 mg/ml, AM7119; Life Technologies) and salmon-sperm DNA (10 mg/ml, 15632-011; Life Technologies) and the mixture was dried in a vacuum concentrator. Hybridization buffer F (20% formamide, 10 mM $NaHPO_4$ at pH 7.0; 50 $\mu$l per hybridization) was added, and the probe/buffer F solution incubated for 3 min at 95°C. Buffer H (4× SSC, 4 mg/ml BSA [acetylated], and 20 mM vanadyl ribonuclease complex; 50 $\mu$l per hybridization) was added in a 1:1 ratio to the probe/buffer F solution. Cells were resuspended in the mixture and incubated overnight at 37°C. After three washes (10% formamide, 2× SSC; 0.1% Triton X-100, 2× SSC; and 1× PBS), cells were resuspended in 1× PBS/DAPI and mounted on glass slides for imaging. Z-planes spaced by 0.2 $\mu$m were acquired on an Ultraview spinning-disc confocal with an Olympus UPlanSAPO 100× objective. Acquisition was carried out with DAPI (405 nm) and FITC (488 nm for alexa488 acquisition) filters.

### FISH data analyses

Images were analysed using ImageJ software (Schneider et al, 2012b). Briefly, Z-stack images (512 × 512 pixels) were generated for DAPI and FISH signal using an average signal intensity stack. Signal intensity was measured across a circle (diameter of 20 pixels) containing the DAPI-stained nucleus or the cytoplasm (DAPI-negative, but inside cell). Average intensity of each circle was calculated and the ratio of nuclear signal/cytoplasmic signal calculated for each cell. Mitotic cells (i.e., two DAPI-stained areas inside the same cell) and cells where the circle did not fit in the cytoplasm were excluded from the analysis. The mean and the confidence interval of the mean were calculated with $\alpha$ = 0.05. Statistical analysis (un-paired *t* test) was carried out with GraphPad Software. Raw data for each sample and the data used to generate the graph in Figs 3B and 4C are listed in Tables S5 and S6.

### RNA sequencing

For spike-in normalisation, *S. cerevisiae* cells were added to *S. pombe* at a 1:10 ratio before RNA isolation. Total RNA was extracted from cultures in mid-log phase using a standard hot phenol method and treated with RNase-free DNase RQ1 (M6101; Promega) to remove DNA. For total RNA sequencing, experiments were performed in duplicates. Ribodepletion was carried out with the ribominus transcriptome isolation kit (K155003; Invitrogen). Poly(A)+ RNA sequencing was performed by using 1/33 of the oligo-d(T) pulldown total volume, subjected to proteinase K treatment for 1 h at 50°C. Poly(A)+ RNA was recovered by a standard hot phenol method. Experiments were performed in triplicate. cDNA libraries were prepared using NEBNext Ultra II Directional RNA Library Prep Kit for Illumina (#E7760S; NEB) for 50 ng of total RNA and using the NEBNext Ultra Directional RNA Library Prep Kit for Illumina (#E7420; NEB) for 100 ng of WT1, *mtl1-1*, *rrp6Δ*, and *dis3-54*-purified oligo-d(T) RNA. Paired-end sequencing was carried out on the Illumina HiSeq 500 platform.

### RNA-sequencing data analyses

Quality trimming of sequenced reads was performed using Trimmomatic (Galaxy Version 0.32.3, RRID:SCR_011848). Reads were aligned to the *S. pombe* genome (ASM294v2.19) using Bowtie 2 (TopHat) (Langmead & Salzberg, 2012). For spike-in normalisation, reads derived from different *S. pombe* and *S. cerevisiae* chromosomes were separated. Reads mapped only once were obtained by SAMTools (Li et al, 2009) and reads were mapped to the genome using genome annotation from Eser et al (2016). Differential expression analyses were performed using DESeq2 (Love et al, 2014) in R and using the spike-in normalisation. For poly(A)+ RNA sequencing, total read count normalisation using DEseq2 (Love et al, 2014) in R was used. The *P*-values indicated below the Venn diagrams were calculated using a standard Fisher's exact test. For GO analysis, AnGeLi (http://bahlerweb.cs.ucl.ac.uk/cgi-bin/GLA/GLA_input), a web-based tool, was used (Bitton et al, 2015).

### Chromatin immunoprecipitation (ChIP) assay

Chromatin immunoprecipitation (ChIP) assays were performed as described (Shah et al, 2014). Immunoprecipitations (IPs) were conducted with antibody against Rpb1 (8WG16, 05-952-I; Millipore) coupled to protein G Dynabeads (10004D; Life Technologies). The values correspond to the Rbp1-ChIP signal relative to the input normalised to a control gene (*fbp1*).

## Data Availability

Raw (fastq) and processed (bedgraph) sequencing data can be downloaded from the National Center for Biotechnology Information (NCBI) Gene Expression Omnibus with the GEO numbers GSE148799 and GSE149187. Mass spectrometry data are available via ProteomeXchange with the identifier PXD016741.

## Supplementary Information

## Acknowledgements

We thank the National Bio Resource Yeast Project, S Grewal, P Bernard and JP Javerzat for strains and constructs. We thank members of the Vasilieva laboratory and Jean-Paul Javerzat for their valuable comments on the manuscript. This work was supported by a Wellcome Trust Senior Research fellowship to L Vasiljeva (WT106994/Z/15/Z), and the Emmy Noether Programme of the Deutsche Forschungsgemeinschaft (DFG) to C Kilchert (KI 1657/2-1). We acknowledge the Micron Advanced Bioimaging Unit (supported by the Wellcome Strategic Awards 091911/B/10/Z and 107457/Z/15/Z) for their support and assistance in performing the FISH experiments.

### Author Contributions

A Birot: conceptualization, investigation, visualization, project administration, and writing—original draft, review, and editing.
K Kus: formal analysis, investigation, and writing—review and editing.
E Priest: investigation.
A Al Alwash: resources and investigation.
A Castello: methodology and writing—review and editing.
S Mohammed: formal analysis, investigation, and methodology.
L Vasiljeva: conceptualization, supervision, funding acquisition, project administration, and writing—original draft, review, and editing.
C Kilchert: conceptualization, investigation, visualization, project administration, and writing—original draft, review, and editing.

### Conflict of Interest Statement

The authors declare that they have no conflict of interest.

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
