## [Reviewer comments · Life Science Alliance]

Life Science Alliance

RNA-binding protein Mub1 and the nuclear RNA exosome act to fine-tune environmental stress response

Adrien Birot, Krzysztof Kus, Emily Priest, Ahmad Al Alwash, Alfredo Castello, Shabaz Mohammed, Lidia Vasiljeva, and Cornelia Kilchert

DOI: <https://doi.org/10.26508/lsa.202101111>

Corresponding author(s): *Cornelia Kilchert, University of Giessen and Lidia Vasiljeva, University of Oxford*

Review Timeline:

Submission Date:	2021-05-03
Editorial Decision:	2021-07-20
Revision Received:	2021-11-09
Editorial Decision:	2021-11-10
Revision Received:	2021-11-12
Accepted:	2021-11-12

Scientific Editor: Novella Guidi

Transaction Report:

July 20, 2021

Re: Life Science Alliance manuscript #LSA-2021-01111-T

Dr. Lidia Vasiljeva
University of Oxford
Biochemistry
South Parks Road
Oxford, Oxfordshire OX13QU
United Kingdom

Dear Dr. Vasiljeva,

Thank you for submitting your manuscript entitled "RNA-binding protein Mub1 and the nuclear RNA exosome act to fine-tune environmental stress response" to Life Science Alliance. The manuscript was assessed by expert reviewers, whose comments are appended to this letter. As you will note from the reviewers' comments below, all reviewers are quite positive and excited about the work that in their views is an interesting study contributing to the RNA exosome field, and should be of interest to all readers with an interest in RNA binding proteins. However, they do raise some minor concerns that need to be addressed in the revised version. We, thus, encourage you to submit a revised version of the manuscript back to LSA that responds to all of the reviewers' points including providing proof that RNA phenotypes are due to defective degradation vs. indirect transcriptional activation by performing experiments suggested by Reviewer 1. Please, also provide quantification of the RNA FISH nucleus/cytoplasm in EV1I-J and explain better the RNA FISH method as requested by Reviewer 3.

Thank you for this interesting contribution to Life Science Alliance. We are looking forward to receiving your revised manuscript.

Sincerely,

B. MANUSCRIPT ORGANIZATION AND FORMATTING:

Reviewer #1 (Comments to the Authors (Required)):

In this paper, Birot and colleagues analyze gene expression in several mutants of the exosome and co-factors in *S.pombe*, and they identify proteins which show increased binding of RNAs in these mutants. They analyze the phenotype of mutants of several of these proteins and then focus on the Mub1 protein. The mub1 mutant shows increased expression of stress responsive genes, and deletion of its zinc finger domain shows a similar phenotype. The authors conclude that Mub1 is required to mediate stress responsive RNAs degradation through the exosome. Overall the studies present an important analysis of gene expression in different exosome mutants, and the discovery of proteins which bind RNA more abundantly in exosome mutants is an interesting finding.

My main concern with the study is the interpretation of the stress induced gene phenotypes. It is unclear whether the increase in hsp70 expression described in Figure 4 is directly due to a lack of degradation of this RNA by the exosome and assisted by Mub1, or because there is an indirect stress response in these mutants, which results in the transcriptional activation of hsp70 - and other responsive genes. Many mutants show indirect stress response effects, and the increased RNA abundance is insufficient to conclude that the effects is direct. Based on the northern analysis shown, there is no way to discriminate transcriptional effects vs. decay and the interpretation that "These findings highlight importance of exosome-dependent mRNA degradation in buffering gene expression networks to mediate cellular adaptation to stress." (copied from the abstract) is not supported by the current data. The authors need to provide direct evidence for degradation of hsp70 (or other stress induced genes) by the exosome and mub1 - for instance by performing half life analysis, or measuring Pol.II occupancy to show that there isn't an indirect transcriptional activation.

Figure 4 also has some issues. There is no evidence of reproducibility for the effects observed. In addition, two different species are detected for hsp70 before heat shock. This is not discussed, and the lower band is likely to correspond to the mature mRNA, the upper band is completely uncharacterized. Furthermore, the double mutant shows an increase in accumulation of the upper product. More work is needed to fully explain and describe the effects shown on this experiment.

Minor points:

Figures in EV1 are shown in the extended version but it would be valuable to include these in the main figures. Especially because figure 4 is so small in terms of content.

I do not think that the polyA nuclear accumulation phenotypes are properly presented. The authors claims that Dis3 mutant shows a milder accumulation of polyA in nucleus than the rrp6 and mtl1 mutants. However, I disagree with this interpretation. There is clear accumulation of polyA in the nucleus in the dis3 mutant, but the pattern is more diffuse in the nucleus/nucleoplasm than for the rrp6 and mtl1 mutant, which show very distinct punctate/foci. These differences in nuclear distribution should be more accurately described.

P5" a high number of mRNAs ...is also increased in the exosome mutants confirming that the nuclear exosome regulates multiple mRNAs in addition to its well-described role in degradation of ncRNAs". Many of these changes in mRNA abundance could result from indirect effects so this conclusion should be moderated.

P8 - the description of SPBC31F10.10c as the homologue of Mub1 and how the authors switched to using the name Mub1 is very confusing:

"We decided to investigate SPBC31F10.10c further, because its deletion led to increased levels of tf2- 1 RNA targeted by the exosome. Additionally, Mub1 shows a >6-fold increase in mtl1-1 RIC (p-value= 5,41E-08) (Figure 2A). SPBC31F10.10c is the homologue of *S. cerevisiae* Mub1".

I would suggest start using the name Mub1 instead of SPBC31F10.10c from the very beginning and just explain that it's because of the homology with *S.cerevisiae*.

Figure 3B. Are Gst2 and SPCC663.08c stress response genes? This should be clarified in the text.

English:

Abstract: In several sentences "the" need to be removed: "mediated by the RNA- binding co-factors"; "is altered in the exosome mutants"; "of a subset of the exosome RNA substrates"

"Transient binding of co-factors either to the exosome or the substrate RNAs as well as their rapid decay" - sentence is misleading as "their rapid decay" seems to be indicate decay of co-factors. Please rephrase.

Reviewer #2 (Comments to the Authors (Required)):

Birot et al. profiled poly(A)⁺ RNAs and their binding proteins (RBPs) in exosome deficient yeast, and compared them with the WT cells. They identified a set of RBPs after stabilization of poly(A)⁺ RNAs, and highlighted their implication in stress response. By deletion of one newly identified RBP, authors confirmed its effects on cell growth upon heat shock.

The study is fruitful with multiple types of data, and the story is written clearly.

This reviewer has only minor comments:

- The authors stated "The underlying hypothesis behind this approach was that stabilization of RNAs targeted by the exosome would facilitate the capture of proteins that are enriched on these RNAs, and that are likely to be functionally linked to the exosome (Figure 1B)." Although it makes sense to discover more RBPs after stabilization of RNAs, they are not logically related to exosome function.
- How to explain the proteins whose RNA-binding abundance were reduced in exosome mutants? Is it because the change of subcellular distribution of RNAs?
- The usage of median-normalization assumes that <50% of protein species are not affected. This assumption may or may not be true in the current study.
- Spiking in *S. cerevisiae* cells for RNAseq is a brilliant design.
- Typos, such as "... a total of total of 1146 ...".

Reviewer #3 (Comments to the Authors (Required)):

The RNA exosome is an key regulator of RNA processing and/or degradation. It matures or degrades almost all the cellular RNAs. The specificity of the complex for its targets relies on cofactors crucial for substrate recognition. Cofactors have been characterized in eukaryotes, but the abundance of different RNA exosome targets indicates that several other cofactors still have to be identified. The authors propose a very elegant approach of poly(A) RNA purification with their associated proteins, aka RNA interactome capture, in the context of RNA exosome depleted cells. In this condition, the RNA exosome sensitive RNAs are stabilized and remain bound to the potential RNA exosome cofactors. They used 3 different mutants affecting the RNA exonuclease activity of the complex (*rrp6Δ*, *dis3-54*) or one of its cofactors (*mtl1-1*). As a proof of principle, the authors confirmed the over accumulation of some of the known RNA targets of the RNA exosome, as well as the increased association of proteins recruiting the RNA exosome on its targets. This approach successfully purified uncharacterized RNA binding proteins with unknown function with potential RNA exosome related functions. They characterized of one this protein, Mub1, confirming its role in poly(A) RNA localization, RNA destabilization and its association with Rrp6, a subunit of the RNA exosome. All these experiments validate their model in which Mub1 is a cofactor of the RNA exosome. The authors provide data showing the importance of the Mub1 Zn finger domain for the protein function. And most interestingly, the authors provided evidences suggesting that Mub1 is involved in the regulation of RNA with a common role in heat shock response.

The authors deliver a very interesting study contributing to the RNA exosome field, and should be of interest to all readers with an interest in RNA binding proteins.

Minor comments:

- p.6: repeat of "in a total of total of..."
- A quantification of the RNA FISH nucleus/cytoplasm in EV11-J (like in 2C) would be interesting to make the point and to compare with 2C
- Fig1C: Is there an explanation why ISS10 is a lot more enriched than Mmi1 in *mtl1-1* and *rrp6Δ* conditions (compare to *dis3-54*), since they are part of the same complex as mentioned in p.4?
- Fig2D: *tf2* RNA is destabilized in *mtl1-1* mutant. Does it mean that *mtl1* has a role in the stabilization of *tf2* RNA in normal conditions? Is it through a different pathway not related to the RNA exosome?
- Since Mub1 and Mtl1 act in the same pathway, is it possible to test their association?
- Is the Mub1 Zn finger domain important for the association with Rrp6? Is the RNA required for this association?
- Fig4B: We can detect 2 bands in lanes 2-3-4 (at 30C), but only the lower band in 5-6-7-8 (at 37C). Are they different temperature-dependent isoforms?
- Including a conclusion section (end of page 9 or top of page 10) would make the paper clearer for the readers.
- p.12: RNA FISH method 5x10⁻¹x10⁸, I guess you meant 5x10⁷-1x10⁸
- p.12: RNA FISH method, objective or magnification used on the confocal?
- p.12: what's the square size used for the FISH quantification (in pixel or μ m)?
- Considering the cell shapes observed in EV3A, how *mub1Δ* phenotype affects your quantification (in 2C) (since the cytoplasm

seems smaller)?

Response to the reviewers' comments

(Reviewers' comments are shown in black and our comments in blue):

We were pleased that the reviewers found our work interesting and we appreciate the opportunity to submit a revised manuscript. The reviewers made excellent suggestions for additional improvements that we have now included. With the addition of the extra data we believe we have strengthened the main conclusions of the manuscript and hope to have satisfied all of the reviewers' concerns.

Thank you for submitting your manuscript entitled "RNA-binding protein Mub1 and the nuclear RNA exosome act to fine-tune environmental stress response" to Life Science Alliance. The manuscript was assessed by expert reviewers, whose comments are appended to this letter. As you will note from the reviewers' comments below, all reviewers are quite positive and excited about the work that in their views is an interesting study contributing to the RNA exosome field, and should be of interest to all readers with an interest in RNA binding proteins. However, they do raise some minor concerns that need to be addressed in the revised version. We, thus, encourage you to submit a revised version of the manuscript back to LSA that responds to all of the reviewers' points including providing proof that RNA phenotypes are due to defective degradation vs. indirect transcriptional activation by performing experiments suggested by Reviewer 1. Please, also provide quantification of the RNA FISH nucleus/cytoplasm in EVII-J and explain better the RNA FISH method as requested by Reviewer 3.

As requested, we now include Pol II ChIP data to demonstrate the absence of transcriptional activation of the stress-responsive genes in the conditions under study (Fig 6A) and quantification of the RNA FISH analysis including detailed documentation (Figs 3B and 4C, and Materials and Methods section). Other points raised by the reviewers are addressed in the point-by-point responses below.

Reviewer #1 (Comments to the Authors (Required)):

In this paper, Birot and colleagues analyze gene expression in several mutants of the exosome and co-factors in *S. pombe*, and they identify proteins which show increased binding of RNAs in these mutants. They analyze the phenotype of mutants of several of these proteins and then focus on the Mub1 protein. The mub1 mutant shows increased expression of stress responsive genes, and deletion of its zinc finger domain shows a similar phenotype. The authors conclude that Mub1 is required to mediate stress responsive RNAs degradation through the exosome. Overall the studies present an important analysis of gene expression in different exosome mutants, and the discovery of proteins which bind RNA more abundantly in exosome mutants is an interesting finding.

We thank the referee for the positive comment.

My main concern with the study is the interpretation of the stress induced gene phenotypes. It is unclear whether the increase in hsp70 expression described in Figure 4 is directly due to a lack of degradation of this RNA by the exosome and assisted by Mub1, or because there is an indirect stress response in these mutants, which results in the transcriptional activation of hsp70 - and other responsive genes. Many mutants show indirect stress response effects, and the increased RNA abundance is insufficient to conclude that the effects is direct. Based on

the northern analysis shown, there is no way to discriminate transcriptional effects vs. decay and the interpretation that "These findings highlight importance of exosome-dependent mRNA degradation in buffering gene expression networks to mediate cellular adaptation to stress." (copied from the abstract) is not supported by the current data. The authors need to provide direct evidence for degradation of *hsp70* (or other stress induced genes) by the exosome and *mub1* - for instance by performing half life analysis, or measuring Pol.II occupancy to show that there isn't an indirect transcriptional activation.

We fully agree with this assessment. To exclude that the stress-responsive transcripts are transcriptionally induced as part of an indirect stress response in the *mub1* mutants, we now include Pol II-ChIP data for the mutant strains (Figure 6A). We did not observe a significant increase in Pol II occupancy over the gene bodies of *gst2*, *SPCC663.08c*, or *hsp16* for *mub1D* or *mub1-DZ-3xFLAG* relative to the control strains, suggesting that the regulation occurs post-transcriptionally rather than through transcriptional activation.

Figure 4 also has some issues. There is no evidence of reproducibility for the effects observed. In addition, two different species are detected for *hsp70* before heat shock. This is not discussed, and the lower band is likely to correspond to the mature mRNA, the upper band is completely uncharacterized. Furthermore, the double mutant shows an increase in accumulation of the upper product. More work is needed to fully explain and describe the effects shown on this experiment.

The upregulation of *hsp16* in *mub1D*, *mtl1-1*, and *mub1D mtl1-1* in the absence of heat shock is very reproducible in our hands and was also detected in the RNA-seq experiment. We now show the distribution of RNA-seq reads across the *hsp16* locus in these strains (Fig S2C) in addition to the read counts (Supplementary table 7). At present, we do not know the identity of the upper bands that we detect with the *hsp16* probe before heat shock. Although the NB probe was not strand-specific, we do not consider an antisense transcript likely, because we did not pick up significant amounts of antisense reads in the RNA-seq under the same conditions. For the time being, our best guess is 3' extended transcripts (the annotated 3'UTR of *hsp16* is comparatively long and may not be present on all transcripts); however, it should be noted that reads in published heat shock RNA-seq data (available on PomBase) span a similar region of the locus as the reads in *mub1D* and *mtl1-1* before heat shock (Rhind et al., 2011). We now include a brief discussion of our interpretation of the band patterns observed at the different temperatures:

"The lower band corresponds to the mature mRNA in size, whereas the upper bands may represent isoforms with longer 5' or 3'UTRs. Please note that the *hsp16* isoforms that are stabilised in the mutants under non-stressed conditions are absent at 37°C, potentially reflecting a higher complexity in *hsp16* regulation, which needs to be further characterised."

Minor points:

Figures in EV1 are shown in the extended version but it would be valuable to include these in the main figures. Especially because figure 4 is so small in terms of content.

The data is now shown as main Figures 2, 3A and B.

I do not think that the polyA nuclear accumulation phenotypes are properly presented. The authors claims that *Dis3* mutant shows a milder accumulation of polyA in nucleus than the *rrp6* and *mtl1* mutants. However, I disagree with this interpretation. There is clear

accumulation of polyA in the nucleus in the *dis3* mutant, but the pattern is more diffuse in the nucleus/nucleoplasm than for the *rrp6* and *mtl1* mutant, which show very distinct punctate/foci. These differences in nuclear distribution should be more accurately described.

Thank you for this observation. We now performed a quantification of the relative FISH signal intensities in nucleus vs. cytoplasm in the exosome mutants (Figure 3B), which confirms the reviewer's view that the *dis3* mutant accumulates significant amounts of poly(A)⁺ RNA in the nucleus, albeit in a more diffuse pattern. We apologize for the misrepresentation. In addition, the differences in poly(A)⁺ RNA accumulation patterns in the different mutants (diffuse vs. punctate) are now addressed in the text (p. 6-7).

P5" a high number of mRNAs ...is also increased in the exosome mutants confirming that the nuclear exosome regulates multiple mRNAs in addition to its well-described role in degradation of ncRNAs". Many of these changes in mRNA abundance could result from indirect effects so this conclusion should be moderated.

We fully agree with this comment and now explicitly refer to this possibility in the text: "..., confirming that the nuclear exosome regulates mRNA levels in addition to its well-described role in degradation of ncRNAs. **However, not all changes necessarily reflect increased mRNA half-lives that are directly related to impaired exosome activity but may be indirect consequences of altered exosome function.**"

P8 - the description of SPBC31F10.10c as the homologue of Mub1 and how the authors switched to using the name Mub1 is very confusing:
"We decided to investigate SPBC31F10.10c further, because its deletion led to increased levels of *tf2-1* RNA targeted by the exosome. Additionally, Mub1 shows a >6-fold increase in *mtl1-1* RIC (p-value= 5,41E-08) (Figure 2A). SPBC31F10.10c is the homologue of *S. cerevisiae* Mub1".

I would suggest start using the name Mub1 instead of SPBC31F10.10c from the very beginning and just explain that it's because of the homology with *S.cerevisiae*.

We now use the gene name Mub1 throughout the manuscript.

Figure 3B. Are *Gst2* and *SPCC663.08c* stress response genes? This should be clarified in the text.

Yes, they are. They were classified as core environmental stress response (CESR) genes in the publication Chen,D., Toone,W.M., Mata,J., Lyne,R., Burns,G., Kivinen,K., Brazma,A., Jones,N. and Bähler,J. (2003) Global Transcriptional Responses of Fission Yeast to Environmental Stress. *Mol. Biol. Cell*, **14**, 214–229, where they are included in the list of "Induced CESR genes" (314 genes, less conservative list using clustering) (http://bahlerweb.cs.ucl.ac.uk/docs/cesr_up_314.pdf), which can be accessed from the supplementary data page (<http://bahlerweb.cs.ucl.ac.uk/projects/stress/>)

We have changed the text to clarify this:

"To validate the RNA-seq data, increased steady-state levels of **two representative CESR mRNAs (69)**, *gst2* and *SPCC663.08c*, in *mub1Δ* and *mtl1-1* mutants were confirmed by Northern blot analyses."

English:

Abstract: In several sentences "the" need to be removed: "mediated by the RNA- binding co-

factors"; "is altered in the exosome mutants"; "of a subset of the exosome RNA substrates" "Transient binding of co-factors either to the exosome or the substrate RNAs as well as their rapid decay" - sentence is misleading as "their rapid decay" seems to be indicate decay of co-factors. Please rephrase.

We went through the manuscript to correct incorrect grammar and inaccurate wording.

Reviewer #2 (Comments to the Authors (Required)):

Birot et al. profiled poly(A)+ RNAs and their binding proteins (RBPs) in exosome deficient yeast, and compared them with the WT cells. They identified a set of RBPs after stabilization of poly(A)+ RNAs, and highlighted their implication in stress response. By deletion of one newly identified RBP, authors confirmed its effects on cell growth upon heat shock. The study is fruitful with multiple types of data, and the story is written clearly.

We thank the referee for the positive evaluation of our work.

This reviewer has only minor comments:

- The authors stated "The underlying hypothesis behind this approach was that stabilization of RNAs targeted by the exosome would facilitate the capture of proteins that are enriched on these RNAs, and that are likely to be functionally linked to the exosome (Figure 1B)." Although it makes sense to discover more RBPs after stabilization of RNAs, they are not logically related to exosome function.

We did not mean to imply that all enriched RBPs are necessarily related to exosome function, but that RBPs related to exosome function are likely to be among the enriched. To clarify this, we changed the sentence to:

"The underlying hypothesis behind this approach was that stabilization of RNAs targeted by the exosome would facilitate the capture of **proteins that are functionally linked to the exosome, which are likely to be enriched on these RNAs** (Figure 1B)."

- How to explain the proteins whose RNA-binding abundance were reduced in exosome mutants? Is it because the change of subcellular distribution of RNAs?

We certainly think that the changed subcellular distribution of RNAs is likely to be a major contributing factor. This is compatible with the data, where we see a strong enrichment of nuclear RBPs vs. proteins involved in translation, which are predominantly localized to the cytoplasm. As we use median-normalization, the analysis is rank-based, so we can only make statements about how the relative extent of poly(A)+ RNA crosslinking between two proteins changed between mutant and WT – a negative value does not necessarily imply a strong reduction in protein crosslinking (also see next point). To emphasize this more strongly, we have changed the description to:

"In contrast, the GO term "cytoplasmic translation" (GO:0002181) was significantly overrepresented among the RBPs **that contributed less to total RNA-protein interactions in the catalytic exosome mutants than in WT**. Indeed, 22 out of 95 RBPs **that are underrepresented in *rrp6Δ*** (p-value = 2,02e-7) and 31 out of 86 RBPs **that are underrepresented in *dis3-54*** (p-value = 7,16e-17) are annotated with "cytoplasmic translation".

- The usage of median-normalization assumes that <50% of protein species are not affected. This assumption may or may not be true in the current study.

Initially, we tried different modes of normalization. In our hands, *rrp6D* and *mtl1-1* pull down significantly more protein per unit of RNA than either WT or *dis3-54* (see also Kilchert et al., MiMB, 2020). We are not sure whether that is because there *are* more RBPs on RNAs in exosome mutants or whether changed metabolism makes them more susceptible to 4-thio-uracil-labelling and crosslinking (ura-inducible genes are among the most affected in exosome mutants according to our RNA-seq data). We settled on the rank-based analysis because we wanted to exclude any such influence. Please note that a different decision would not have affected the list of candidate proteins chosen for further analysis.

- Spiking in *S. cerevisiae* cells for RNAseq is a brilliant design.

- Typos, such as "... a total of total of 1146 ...".

Has been corrected.

Reviewer #3 (Comments to the Authors (Required)):

The RNA exosome is a key regulator of RNA processing and/or degradation. It matures or degrades almost all the cellular RNAs. The specificity of the complex for its targets relies on cofactors crucial for substrate recognition. Cofactors have been characterized in eukaryotes, but the abundance of different RNA exosome targets indicates that several other cofactors still have to be identified. The authors propose a very elegant approach of poly(A) RNA purification with their associated proteins, aka RNA interactome capture, in the context of RNA exosome depleted cells. In this condition, the RNA exosome sensitive RNAs are stabilized and remain bound to the potential RNA exosome cofactors. They used 3 different mutants affecting the RNA exonuclease activity of the complex (*rrp6Δ*, *dis3-54*) or one of its cofactors (*mtl1-1*). As a proof of principle, the authors confirmed the over accumulation of some of the known RNA targets of the RNA exosome, as well as the increased association of proteins recruiting the RNA exosome on its targets. This approach successfully purified uncharacterized RNA binding proteins with unknown function with potential RNA exosome related functions. They characterized one of these proteins, Mub1, confirming its role in poly(A) RNA localization, RNA destabilization and its association with Rrp6, a subunit of the RNA exosome. All these experiments validate their model in which Mub1 is a cofactor of the RNA exosome. The authors provide data showing the importance of the Mub1 Zn finger domain for the protein function. And most interestingly, the authors provided evidences suggesting that Mub1 is involved in the regulation of RNA with a common role in heat shock response. The authors deliver a very interesting study contributing to the RNA exosome field, and should be of interest to all readers with an interest in RNA binding proteins.

We thank the referee for their positive evaluation of the significance of our observations.

Minor comments:

-p.6: repeat of "in a total of total of...

Has been corrected.

-A quantification of the RNA FISH nucleus/cytoplasm in EV11-J (like in 2C) would be interesting to make the point and to compare with 2C

FISH signal quantification is now included for all FISH experiments (Figures 3B and 4C).

-Fig1C: Is there an explanation why ISS10 is a lot more enriched than Mmi1 in *mtl1-1* and *rrp6Δ* conditions (compare to *dis3-54*), since they are part of the same complex as mentioned in p.4?

This is an interesting observation. It is certain that Mmi1 is the initiating protein that recognizes meiotic RNAs through direct binding and sets off the exosome-dependent degradation cascade. We do not know whether it does this as part of a pre-assembled complex or whether the other proteins assemble once the target has been selected – if the latter was the case, Mmi1 would have a longer residence time on the RNA in WT than proteins that are recruited later, and the differential enrichment could reflect these kinetics. Also, Iss10 is an important component of nuclear exosome foci (Yamashita et al., NAR 2012; Egan et al., RNA 2014) – these may or may not correspond to the foci where we see poly(A)⁺ RNA accumulate in *rrp6Δ* and *mtl1-1*, and crosslinking to Iss10 may be favored in these compartments. However, we cannot exclude that differences in mass spec-ability contribute to the differential enrichment that we observe for the two proteins.

-Fig2D: *tf2* RNA is destabilized in *mtl1-1* mutant. Does it mean that *mtl1* has a role in the stabilization of *tf2* RNA in normal conditions? Is it through a different pathway not related to the RNA exosome?

The decrease in *tf2-1* levels in *mtl1-1* is reproducible in Northern blots and is also observed in the RNA-seq data. Remarkably, *mtl1-1* also rescues the *mub1D* phenotype (see below). We do not fully understand this phenotype. However, many transposable elements lie within HOODs (heterochromatin domains), which show RNAi-dependent hyper-heterochromatinization when *rrp6* is deleted (Lee et al., Cell 2013). Several Mtl1-interacting proteins are required for HOOD formation, although a direct involvement of Mtl1 has not been tested. At present, we consider it most likely that *mtl1-1* affects *tf2-1* levels by altering H3K9me levels across the gene, but we have not looked into this any further.

-Since Mub1 and Mtl1 act in the same pathway, is it possible to test their association?

The association of Mub1 with the nuclear RNA exosome was a relevant point to address. We chose Rrp6 for the co-IP because it is specific to the nuclear exosome complex and directly involved in RNA degradation. However, we also did a Mub1-3XFLAG purification from cycling cells at 30°C followed by MS-MS analysis. Mtl1 was mildly enriched in the purification fraction (5 peptides, intensity, 8,54E+07) compared to an untagged control strain (1 peptide, intensity 6,42E+06), but not strongly enough to support a direct interaction between the two proteins (data not included in the manuscript).

-Is the Mub1 Zn finger domain important for the association with Rrp6? Is the RNA required for this association?

To test whether the presence of RNA is required for the interaction between Mub1 and Rrp6, we performed co-immunoprecipitation with or without addition of RNase A. RNase A treatment abolished the Mub1-Rrp6 interaction. This data is now included in the manuscript (Figure 6C).

-Fig4B: We can detect 2 bands in lanes 2-3-4 (at 30C), but only the lower band in 5-6-7-8 (at 37C). Are they different temperature-dependent isoforms?

As outlined in our response to reviewer 1 who also commented on the second band in the *hsp16* Northern blot, which is only observed at low temperature, we do not know the identity of the upper band. Because we did not pick up significant amounts of antisense reads in the RNA-seq under the same conditions, we consider an antisense transcript unlikely. We think it most likely that this band represents a temperature-dependent 5' or 3'-extended isoform, as the reviewer suggests; however, we have not characterized this band any further. To illustrate the uncertainty, we now include a brief discussion of our interpretation of the band patterns observed at the different temperatures:
“The lower band corresponds to the mature mRNA in size, whereas the upper bands may represent isoforms with longer 5' or 3'UTRs. Please note that the *hsp16* isoforms that are stabilised in the mutants under non-stressed conditions are absent at 37°C, potentially reflecting a higher complexity in *hsp16* regulation, which needs to be further characterised.”

-Including a conclusion section (end of page 9 or top of page 10) would make the paper clearer for the readers.

We now present the general conclusion of our work in a distinct Discussion section (p. 10).

-p.12: RNA FISH method 5x10-1x108, I guess you meant 5x10⁷-1x10⁸

Has been corrected.

-p.12: RNA FISH method, objective or magnification used on the confocal?

The objective was an Olympus UPlanSAPO with a 100X magnification. This information is now included in the Materials and Methods section:
“Z-planes spaced by 0.2 μm were acquired on a Ultraview spinning-disc confocal with an Olympus UPlanSAPO 100x objective.”

-p.12: what's the square size used for the FISH quantification (in pixel or um)?

We used a circle with a diameter of 20 pixels. We now include a section “FISH data analysis” in the Materials and Methods section where we describe the FISH quantification in detail.

-Considering the cell shapes observed in EV3A, how *mub1Δ* phenotype affects your quantification (in 2C) (since the cytoplasm seems smaller)?

The initial calibration was performed on WT images to determine the optimal size of the circle to make the measurement across the different mutants. As the reviewer points out, the rounded cell shape characteristic of *mub1Δ* cells sometimes made it difficult to select an area in the cytoplasm for quantification. Cells where this was not possible were excluded from the analysis. This information is now included in the section “FISH data analysis”:
“Mitotic cells (i.e., two DAPI-stained areas inside the same cell) and cells where the circle did not fit in the cytoplasm were excluded from the analysis.”

November 10, 2021

RE: Life Science Alliance Manuscript #LSA-2021-01111-TR

Dr. Cornelia Kilchert
Justus-Liebig University Giessen
Institute of Biochemistry
Giessen
Germany

Dear Dr. Kilchert,

Thank you for submitting your revised manuscript entitled "RNA-binding protein Mub1 and the nuclear RNA exosome act to fine-tune environmental stress response". We would be happy to publish your paper in Life Science Alliance pending final revisions necessary to meet our formatting guidelines.

- please add ORCID ID for the corresponding author-you should have received instructions on how to do so
- please separate the Figure legends and Supplemental Figure legends into separate sections

A. FINAL FILES:

B. MANUSCRIPT ORGANIZATION AND FORMATTING:

Sincerely,

November 12, 2021

RE: Life Science Alliance Manuscript #LSA-2021-01111-TRR

Dr. Cornelia Kilchert
University of Giessen
Institute of Biochemistry
Giessen
Germany

Dear Dr. Kilchert,

Thank you for submitting your Research Article entitled "RNA-binding protein Mub1 and the nuclear RNA exosome act to fine-tune environmental stress response". It is a pleasure to let you know that your manuscript is now accepted for publication in Life Science Alliance. Congratulations on this interesting work.

DISTRIBUTION OF MATERIALS:

Again, congratulations on a very nice paper. I hope you found the review process to be constructive and are pleased with how the manuscript was handled editorially. We look forward to future exciting submissions from your lab.

Sincerely,
